# Factors Associated with Evidence-Based Practice Competencies among Taiwanese Nurses: A Cross-Sectional Study

**DOI:** 10.3390/healthcare12090906

**Published:** 2024-04-26

**Authors:** Li-Chuan Cheng, Chia-Jung Chen, Shih-Chun Lin, Malcolm Koo

**Affiliations:** 1Department of Nursing, Dalin Tzu Chi Hospital, Buddhist Tzu Chi Medical Foundation, Chiayi 62247, Taiwan; 2School of Nursing, College of Nursing, National Taipei University of Nursing and Health Sciences, Taipei City 112303, Taiwan; 3Department of Nursing, College of Nursing, Tzu Chi University of Science and Technology, Hualien 970302, Taiwan; 4Dalla Lana School of Public Health, University of Toronto, Toronto, ON M5T 3M7, Canada

**Keywords:** evidence-based practice, cross-sectional survey, nurses, nurse practitioners, Taiwan

## Abstract

Evidence-based practice (EBP) is an essential component of healthcare practice that ensures the delivery of high-quality care by integrating the best available evidence. This study aimed to explore factors influencing EBP among nursing professionals in Taiwan. A cross-sectional survey study was conducted with 752 registered nurses and nurse practitioners recruited from a regional teaching hospital in southern Taiwan. EBP competency was evaluated using the Taipei Evidence-Based Practice Questionnaire (TEBPQ). The results showed that participation in evidence-based courses or training within the past year had the strongest association with EBP competencies (Std. B = 0.157, *p* < 0.001). Holding a graduate degree (Std. B = 0.151, *p* < 0.001), working in gynecology or pediatrics (Std. B = 0.126, *p* < 0.001), searching the literature in electronic databases (Std. B = 0.072, *p* = 0.039), and able to read academic articles in English (Std. B = 0.088, *p* = 0.005) were significantly associated with higher TEBPQ scores. Younger age (Std. B = −0.105, *p* = 0.005) and male gender (Std. B = 0.089, *p* = 0.010) were also identified as factors contributing to higher EBP competencies. The study highlights the importance of ongoing professional development, including EBP training and language proficiency, in enhancing EBP competencies among nursing professionals in Taiwan.

## 1. Introduction

Evidence-based practice (EBP) is an essential component of contemporary healthcare practice, ensuring the delivery of high-quality and efficient patient care [1]. It involves judiciously integrating current best research evidence with clinical expertise and patient values to facilitate informed clinical decisions [2]. Despite its recognized importance, the adoption and implementation of EBP among healthcare professionals vary widely [3,4]. This inconsistency can be attributed to various factors, including but not limited to a lack of resources, time constraints, and varying levels of EBP competencies [5,6]. Competency encompasses the intricate attributes of knowledge, skills, and attitudes necessary for efficiently and effectively performing a set of tasks to an appropriate standard [7]. EBP competencies involve fundamental skills required to engage in evidence-based practices, such as formulating clinical questions, conducting efficient literature searches, critically appraising evidence, and applying findings to clinical decision-making.

In the context of nursing care, incorporating EBP into nursing education and practice is crucial for advancing nursing science, enhancing nursing care, and improving patient outcomes [8,9]. By combining the best available evidence with clinical experience and patient preferences, nurses are equipped to make well-informed decisions that optimize patient care. This approach enables the development of care protocols that are not only based on the latest and most reliable research but also address the unique needs of individual patients. As nurses continually adapt their practices to reflect new evidence, they can sustain ongoing professional development and quality improvement in healthcare settings [10].

Previous research has identified various factors influencing EBP competencies. For instance, a cross-sectional study among 185 nurses in a Norwegian cancer hospital found that those educated in EBP and participating in EBP working groups exhibited stronger EBP beliefs compared to their peers who had not [11]. Another cross-sectional survey involving 491 nurses and 78 allied healthcare providers in Switzerland showed higher EBP implementation levels among those with formal training and those occupying senior professional roles [12]. Similarly, an institutional-based cross-sectional study in Ethiopia involving a random sample of 418 nurses showed that male sex, work experience of more than five years, a head nurse role in the hospital, master’s degree education, and the availability of evidence-based nursing practice guidelines were significant independent factors associated with EBP utilization [13]. A recent global scoping review encompassing 47 studies revealed that educational attainment, participation in EBP education, research involvement, and organizational support are key contributors to EBP knowledge and skills among nurses [14]. 

Despite extensive research on the factors influencing EBP competency, prior studies have primarily used instruments focusing on specific aspects of EBP, such as knowledge, attitudes, or skills, and were developed based on Western contexts. There is a need for further research that covers a broader range of dimensions, including attitudes toward EBP, question formulation abilities, evidence searching, literature appraisal, and the application of findings in clinical practice. This study used the Taipei Evidence-Based Practice Questionnaire (TEBPQ), which was developed because of the recognized need for a succinct tool to assess healthcare professionals’ understanding and application of EBP principles in clinical settings [15]. The TEBPQ defines the EBP essential skills across five distinct domains: Ask, Acquire, Appraisal, Apply, and Attitude. These categories represent a systematic approach that mirrors the four-step EBP model at the clinical level, which involves formulating clinically relevant questions (Ask), gathering evidence (Acquire), evaluating its validity and applicability (Appraisal), and utilizing this evidence in clinical practice (Apply). In addition, the Attitude domain explores the psychological factors influencing EBP, evaluating practitioners’ readiness and their perceived value of EBP. This study aimed to investigate the factors associated with EBP competencies in the Taiwanese healthcare context using the TEBPQ. A cross-sectional study design was used for this broader exploration to assess these diverse competencies and to provide an overview of the current state of EBP competencies in a specific healthcare setting.

## 2. Materials and Methods

### 2.1. Study Design and Data Collection

A cross-sectional questionnaire survey was conducted from 15 February to 31 March 2023, targeting registered nurses and nurse practitioners at a regional teaching hospital in southern Taiwan. The hospital, a private hospital affiliated with the Buddhist Tzu Chi Foundation, is accredited by the Joint Commission of Taiwan. This accreditation body in healthcare was founded by the former Department of Health (renamed to the Ministry of Health and Welfare in 2013), the Taiwan Hospital Association, the Taiwan Non-Government Hospitals and Clinics Association, and the Taiwan Medical Association. The hospital has more than 500 beds and serves as a primary clinical education site for nursing students in the region.

A purposive sampling strategy was used to recruit participants. First, the head nurses from various units within the study hospital were approached and provided with a detailed explanation of the study. Following these briefings, they were given a number of survey packs, each containing a self-administered questionnaire, a consent form, and a convenience store gift card valued at TWD 100 (approximately USD 3.3). The head nurses then distributed these survey packs to the nurses in their respective units. Nurses who consented to participate completed the questionnaire and kept the gift card. Those who chose not to participate returned the uncompleted questionnaire with the gift card. Regardless of their decision, all were instructed to return the sealed survey packs to the head nurses. The researcher made regular visits to each unit to collect the returned survey packs from the head nurses.

### 2.2. Ethical Considerations

The survey packs were designed to be sealed by participants after completion, ensuring that their responses remained confidential and were only accessible to the research team. Upon collection, the questionnaire and consent forms were immediately stored separately to safeguard participant anonymity further. Each questionnaire was assigned a unique code, and to enhance privacy, no personal identifiers were included on any of the questionnaires. The study protocol received approval from the Research Ethics Committee of Dalin Tzu Chi Hospital, Buddhist Tzu Chi Medical Foundation (No. B11102009) (Approval date: 11 May 2022).

### 2.3. Instruments

A self-administered paper-based questionnaire was used to ascertain information on participants’ demographics, work characteristics, previous experience in EBP training, and effectiveness of EBP education. EBP competency was assessed using the TEBPQ. Permission to use the TEBPQ was obtained from the scale developer, Dr. Kee-Hsin Chen. TEBPQ consists of 26 items, divided into five domains: “Ask” (5 items), “Acquire” (7 items), “Appraisal” (4 items), “Apply” (6 items), and “Attitude” (4 items) of evidence-based practice, on a 5-point Likert scale from 1 (strongly disagree) to 5 (strongly agree). The psychometric properties of the TEBPQ were evaluated in a study consisting of content validity evaluation by a panel of experts and construct validity on 136 Taiwanese participants. The results showed that the content validity index of TEBPQ was 0.9 with an internal reliability Cronbach’s α of 0.87. A contrasted-group approach for construct validity showed that all *p* values were significant among the five domains, indicating that TEBPQ is an easy-to-use instrument with good validity and reliability for evaluating the effectiveness of EBP education [15]. 

### 2.4. Data Analysis

The collected data were analyzed using IBM SPSS Statistics for Windows, version 29.0 (IBM Inc., Armonk, NY, USA). Continuous variables were summarized using means and standard deviations (SD), while categorical variables were presented as frequencies and percentages. Simple linear regression analysis was performed to identify factors associated with the TEBPQ scores. Variables with a *p*-value of < 0.20 in the univariate analysis were subjected to further evaluation in a multiple linear regression model, adopting a stepwise selection procedure. The presence of multicollinearity among the independent variables was evaluated using the variance inflation factor. A two-tailed *p*-value of <0.05 was considered statistically significant.

## 3. Results

Of the 780 questionnaires distributed, 752 were completed, representing a response rate of 96.4%. Table 1 outlines the basic characteristics of the 752 participants in this study, who were predominantly female (n = 704, 93.6%) with a mean age of 34.9 years (SD = 9.39). These participants reported a mean of 12.4 years (SD = 8.60) of nursing experience. The majority were registered nurses (n = 646, 85.9%), and 7% held management-level positions.

The participants worked across various units: 18.0% in internal medicine, 14.9% in surgery, 4.1% in psychosomatic medicine, 4.7% in gynecology or pediatrics, 23.9% in critical care, and 34.4% in non-ward units. Most held an associate’s or bachelor’s degree (92.4%), and 7.6% possessed a graduate degree. Regarding marital status, 63.4% were single or unmarried, and 68.8% had no children. The professional distribution was as follows: 15.0% were either N or NP; 67.0% were NP2, N1, or N2; 13.6% were N3 or NP3; and 4.4% were NP4, NP5, or N4.

Table 2 summarizes the participants’ previous experience in evidence-based training. Half of them (50.4%) had not attended any evidence-based courses or training in the past year, while 41.0% had attended one to two times, primarily motivated by work requirements. Engagement with the literature in electronic databases varied, with 46.5% doing so several times a year. The dissemination of EBP through academic outputs was low, with 91.5% having never published evidence-based academic posters and 95.7% having never published evidence-based journal articles. Moreover, 82.2% reported inadequate or highly inadequate ability to read academic articles in English.

The mean TEBPQ score among participants was 86.8 (SD = 15.5), with scores ranging from 26 to 130. Figure 1 illustrates the distribution of Likert responses for the 26 TEBPQ items using a diverging stacked bar graph. Each item’s responses are divided into five categories: strongly disagree (red), disagree (orange), neutral (blue), agree (light green), and strongly agree (dark green). The graph, centered around a neutral zero point, shows that most responses agreed or strongly agreed with the survey items, as evidenced by the prominent light and dark green bars extending to the right.


**Domain**

**Items**
Ask
Q1.I am able to construct background questions.Q2.I am able to construct answerable questions using PICO (patient/problem, intervention/indicator, comparator, and outcome).Q3.I am able to differentiate the types of clinical questions, e.g., therapies, etiology/ harm, diagnosis, prognosis/prevention…etc.Q4.I am able to raise questions constantly in my daily work.Q5.I am able to record clinical questions for later answering.
Acquire
Q6.I am able to define appropriate keywords for searching.Q7.I know the best sources of current evidence for my clinical discipline.Q8.I know how to find the best evidence to solve my clinical questions.Q9.I am able to find the best evidence in 15 min.Q10.I am able to use more than one database for widening the scope of information.Q11.I am able to use the advanced function of search engine.Q12.I am able to save keywords and searching strategies for future updating
Appraisal
Q13.I understand the commonly used terms in evidence-based medicine, e.g., randomized controlled trial (RCT), number needed to treat (NNT)…etc.Q14.I am able to understand “level of evidence” of a paper.Q15.I am able to appraise literature critically.Q16.I am able to create appraisal summaries, e.g., using Question Log or CATmaker …etc.
Apply
Q17.I am able to apply literature evidence to my clinical practice.Q18.I can reiterate evidence as plain language for patients.Q19.I am able to make appropriate decision while clinical experiences are different from literature evidence.Q20.I am able to evaluate clinical outcomes by evidence-based quality indicators.Q21.I am able to integrate 3 “E”s for clinical decision making. (3 “E”s: evidence, expertise and expectation).Q22.I am able to apply evidence-based clinical guidelines in healthcare.
Attitude
Q23.I think the concept of evidence-based practice (EBP) has been emphasized in clinical settings.Q24.I think clinical professionals should have knowledge and skill of EBP.Q25.I think EBP can prevent healthcare disputes.Q26.I think EBP competencies have helped significantly in my practice.


Table 3 presents findings from simple and multiple stepwise linear regression analyses, identifying factors associated with TEBPQ scores. The simple linear regression analysis revealed 14 variables significantly associated with TEBPQ scores across demographic, educational, and professional experience categories. However, the multiple stepwise linear regression analysis showed a slightly different pattern, with ten variables independently associated with TEBPQ scores.

Age was inversely associated with TEBPQ scores (Std. B = −0.105, *p* = 0.005), suggesting that younger nursing professionals aligned more with EBP. Being male was significantly associated with higher TEBPQ scores (Std. B = 0.089, *p* = 0.010), as was being a registered nurse compared to a nurse practitioner (Std. B = 0.088, *p* = 0.015). Administrative roles were also significantly associated with higher TEBPQ scores (Std. B = 0.080, *p* = 0.030), as were working in gynecology or pediatrics units (Std. B = 0.126, *p* < 0.001) compared to other units (Std. B = 0.126, *p* < 0.001). Participants with a graduate degree (Std. B = 0.151, *p* < 0.001) and those who had attended evidence-based courses or training in the past year (Std. B = 0.157, *p* < 0.001) were significantly associated with higher TEBPQ scores. Moreover, searching for the literature in electronic databases (Std. B = 0.114, *p* < 0.001) and publishing evidence-based academic posters three or more times (Std. B = 0.072, *p* = 0.039) were significantly associated with higher TEBPQ scores. Lastly, the ability to read academic articles in English was associated with higher TEBPQ scores (Std. B = 0.088, *p* = 0.005).

## 4. Discussion

The EBP competency of 752 nurses from a regional teaching hospital in southern Taiwan was evaluated in a cross-sectional survey study using the TEBPQ. Ten factors significantly influencing EBP competencies were identified. Of them, the strongest association, as indicated by the magnitude of the standardized regression coefficients, was observed for those who had attended evidence-based courses or training in the past year. This finding is consistent with previous studies that stated that participation in EBP education was a significant factor associated with EBP competencies among nurses [14,16]. A randomized controlled trial showed that EBP educational intervention had a positive effect on emergency nurses’ EBP attitudes, knowledge, self-efficacy, skills, and behavior, which was most apparent six months after the education but decreased to baseline level afterward [17]. A scoping review of 47 studies identified EBP education as a key factor in elevating nurses’ knowledge and skills [14].

Second, possessing a graduate degree was significantly associated with higher TEBPQ scores. This finding contrasts a study of 361 Taiwanese nurses, where those with a master’s or higher degree had significantly higher EBP competency [16]. Another study of 473 registered nurses working in 10 hospitals in the Greek also found that having a master’s degree was significantly associated with the domains of attitudes, knowledge, and skills of EBP [18].

Third, compared to working in other units, working in the gynecology or pediatrics unit was associated with significantly higher TEBPQ scores. Gynecology, which focuses on maternal health during pregnancy, childbirth, and the postpartum period, along with pediatrics, covering a wide age range from newborns to adolescents, requires specialized care due to patients’ distinct developmental stages and unique health needs. As a result, healthcare workers in these units are facing increased risks of legal claims and indemnity payments [19,20]. It is hypothesized that this elevated risk might enhance the incentive among providers in these units to develop a heightened awareness and appreciation for EBP, potentially leading to stronger beliefs and more robust implementation of EBP. However, this hypothesis is preliminary, and further research is needed to explore these dynamics in a broader healthcare context.

Fourth, searching for literature in electronic databases emerged as another significant factor associated with higher TEBPQ scores. The ability to search for literature in electronic databases is a core skill of EBP and is part of the items in the TEBPQ, thus reflecting higher scores. Engagement with research databases may indicate a proactive approach to professional development closely associated with EBP, potentially leading to higher TEBPQ scores. However, further research is needed to substantiate this hypothesis, possibly by examining the direct impacts of various database use on TEBPQ scores or exploring longitudinal relationships between database usage and ongoing professional development in EBP.

Fifth, younger age was associated with significantly higher TEBPQ scores. A systematic analysis of knowledge, attitudes, implementation, facilitators, and barriers of EBP in community nurses revealed that younger age, academic training, and management functions are the most cited facilitators of positive attitudes towards EBP [5]. It is hypothesized that younger nurses are likely to have graduated more recently and may have been exposed to current EBP curricula, including EBP, leading to higher TEBPQ scores. In addition, it is possible that early-career professionals may engage more actively with EBP activities, workshops, and continued education, which can positively impact their TEBPQ scores. This conjecture about age-related disparities in EBP attitudes will require further research to confirm.

Sixth, male nursing professionals reported significantly higher TEBPQ scores than their female counterparts. This finding is consistent with a study involving 188 nursing bridge program students at a university in Saudi Arabia, where EBP beliefs were significantly associated with male sex, among other factors [21]. A cross-sectional study of 418 nurses in Ethiopia showed that male nurses were 4.65 times more likely to utilize EBNP compared to their female counterparts. The authors suggested that the reason is that female nurses are busy due to heavier familial and social duties [13]. Other studies suggested that gender norms and expectations can influence attitudes towards EBP [22]. An integrative review study of workplace gender discrimination towards registered nurses found that male nurses perceived themselves as more diligent, rational, decisive, independent, technically skilled, and possessing superior management and expertise compared to female colleagues [23]. These findings suggest the influence of gender norms and expectations on professional self-perception and behavior. Therefore, it is reasonable to suggest that these gendered perceptions and societal expectations could shape attitudes toward EBP, potentially predisposing male nurses to embrace EBP principles more readily due to their self-perceived competencies and leadership qualities.

Seventh, having an adequate ability to read academic articles in English was a significant factor associated with TEBPQ scores. This finding is logical because proficiency in English reading skills is crucial for accessing and understanding the research literature, which forms the foundation of EBP. A qualitative study of 64 Saudi Arabian nurses also revealed that a lack of references in their native language and the inability to distinguish between high-quality research studies were barriers to EBP application [24].

Eighth, registered nurses reported significantly higher TEBPQ scores than their nurse practitioners. This may be explained by the fact that registered nurses often have a broader range of responsibilities that can encompass many aspects of patient care, which may necessitate a wider application of evidence-based practice. Their roles may require them to stay current with various clinical practices and standards, prompting them to engage more frequently with the latest research and evidence-based guidelines.

Ninth, being at the management level was a significant factor associated with higher TEBPQ scores. As leaders within hospital settings, nurses at the management level were often expected to promote a culture of EBP, facilitating staff training and allocating resources to support the implementation of evidence-based interventions. Their higher EBP scores may stem from a deeper understanding of the importance of EBP in improving patient outcomes, enhancing quality of care, and driving organizational success.

Tenth, publishing evidence-based academic posters three times or more was associated with significantly higher TEBPQ scores. Creating poster presentations requires the synthesis and application of EBP principles, a skill fundamental to EBP and likely to contribute to higher TEBPQ scores. Moreover, publishing posters usually means actively engaging with the latest research, which may reinforce the skills and knowledge assessed by the TEBPQ.

Several limitations are worth mentioning. First, the survey was conducted at a single regional teaching hospital in southern Taiwan, which limited its generalizability. Future research should consider a broader, more diverse sample to validate these findings and explore additional factors influencing EBP competency among Taiwanese nursing professionals. Second, the assessment of TEBPQ relied on self-reported data, which could introduce bias. Specifically, social desirability bias might lead participants to report a more favorable agreement with EBP than they actually hold. Future research should expand to a broader, more diverse sample to validate these findings.

## 5. Conclusions

The findings of this study advance our understanding of EBP by integrating demographic and professional variables with EBP competencies as quantified through the TEBPQ. Our findings indicate that younger professionals, males, registered nurses, and those in administrative roles exhibit higher EBP competencies, suggesting a generational shift and role-specific engagement in EBP. This could reflect differences in resource access, responsibilities, or professional cultures.

On a practical level, this study supports the importance of targeted professional development and strategic resource allocation to enhance EBP competencies. The correlations between EBP training, proficiency in English, and database usage with higher competency scores also support the need for continuous professional development and more focused resource distribution, ensuring all staff is equipped for high-quality, evidence-based care.

## Figures and Tables

**Figure 1 healthcare-12-00906-f001:**
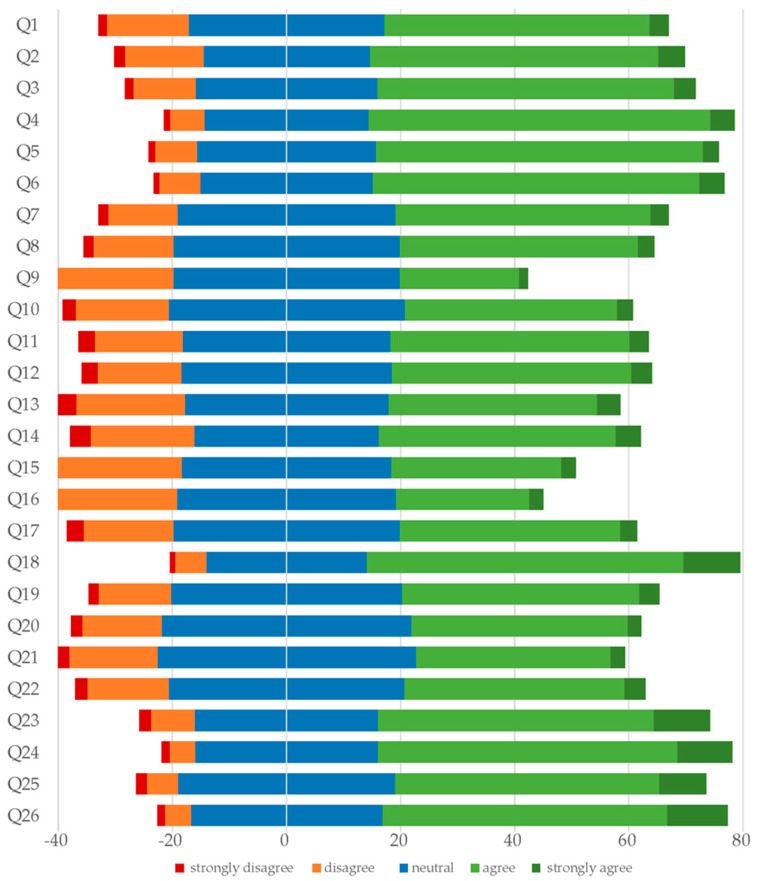
Diverging stacked bar graph showing the distribution of Likert responses for the Taipei Evidence-Based Practice Questionnaire (TEBPQ).

**Table 1 healthcare-12-00906-t001:** Basic characteristics of the study participants (N = 752).

Variable	N (%)
Age, mean (standard deviation)	34.9 (9.39)
Sex	
Male	48 (6.4)
Female	704 (93.6)
Nursing experience, years, mean (standard deviation)	12.4 (8.60)
Job title	
Registered nurses	646 (85.9)
Nurse practitioners	106 (14.1)
Role type	
Management level	53 (7.0)
Non-management level	699 (93.0)
Work unit	
Internal medicine	135 (18.0)
Surgery	112 (14.9)
Psychosomatic Medicine	31 (4.1)
Gynecology or pediatrics	35 (4.7)
Critical care	180 (23.9)
Non-ward unit	259 (34.4)
Educational level	
Associate’s or bachelor’s degrees	695 (92.4)
Graduate-degree	57 (7.6)
Marital status	
Single or unmarried	477 (63.4)
Other (being married, divorced, widowed)	275 (36.6)
Number of children	
0	517 (68.8)
1	72 (9.6)
2	136 (18.1)
≥3	27 (3.6)
Professional level	
N or NP	113 (15.0)
NP2 or N1 or N2	504 (67.0)
N3 or NP3	102 (13.6)
NP4 or NP5 or N4	33 (4.4)

**Table 2 healthcare-12-00906-t002:** Previous experience in evidence-based training.

Variable	N (%)
Evidence-based courses or training attended in the past year	
0 times	379 (50.4)
1–2 times	308 (41.0)
≥3 times	65 (8.6)
Motivation for attending evidence-based courses or training (multiple answer question, top five combinations from five choices)	
Work requirement	425 (56.5)
Assigned by organization	94 (12.5)
Professional development	70 (9.3)
School work requirement	38 (5.1)
Work requirements and assigned by the organization	34 (4.5)
Other combinations	91 (12.1)
Frequency of searching for literature in electronic databases	
≥1/day	59 (7.8)
Several times/week	72 (9.6)
Several times/month	214 (28.5)
Several times/year	350 (46.5)
Not used	57 (7.6)
Had published evidence-based academic posters	
Never	688 (91.5)
1–2 times	61 (8.1)
≥3 times	3 (0.4)
Had published evidence-based journal articles	
Never	720 (95.7)
1–2 times	27 (3.6)
≥3 times	5 (0.7)
Ability to read academic articles in English	
Highly adequate	65 (8.6)
Adequate	69 (9.2)
Inadequate	313 (41.6)
Highly inadequate	305 (40.6)

**Table 3 healthcare-12-00906-t003:** Simple and multiple stepwise linear regression analysis of factors associated with Taipei Evidence-Based Practice Questionnaire (TEBPQ) scores.

Variable	Simple Linear Regression		Multiple Linear Regression	
	Std. B	*p*	Std. B	*p*
Age (year)	−0.090	0.013	−0.105	0.005
Male (ref. = female)	0.109	0.003	0.089	0.010
Nursing experience, years	−0.081	0.027	–	–
Job title (ref. = nurse practitioners)	0.065	0.074	0.088	0.015
Role type (ref. = non-administrative)	0.147	<0.001	0.080	0.030
Work unit (ref. = internal medicine)				
Surgery	0.049	0.277	–	–
Psychosomatic Medicine	0.041	0.291	–	–
Gynecology or pediatrics	0.161	<0.001	0.126	<0.001
Critical care	0.122	0.011	–	–
Non-ward Unit	0.038	0.446	–	–
Educational level (ref. = associate’s or bachelor’s degrees)	0.196	<0.001	0.151	<0.001
Marital status (ref. = unmarried)	0.061	0.093	–	–
Number of children (ref. = 0)				
1	−0.006	0.868	–	–
2	−0.035	0.352	–	–
≥3	0.011	0.771	–	–
Nurse level (ref. = N or NP)				
NP2 or N1 or N2	0.013	0.786	–	–
N3 or NP3	0.091	0.051	–	–
NP4 or NP5 or N4	0.083	0.040	–	–
Evidence-based courses or training attended in the past year (ref. = none)	0.224	<0.001	0.157	<0.001
Searched for literature in electronic databases (ref. = never)	0.162	<0.001	0.114	<0.001
Had published evidence-based academic posters (ref. = never)				
1–2 times	0.086	0.018	–	–
≥3 times	0.120	<0.001	0.072	0.039
Had published evidence-based articles in journals (ref. = never)				
1–2 times	0.107	0.003	–	–
≥3 times	0.030	0.404	–	–
Ability to read academic articles in English (ref. = highly inadequate and inadequate)	0.095	0.009	0.088	0.005

## Data Availability

The data presented in this study are available on request from the corresponding author.

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
