# Peer review of "Factors Associated with Evidence-Based Practice Competencies among Taiwanese Nurses: A Cross-Sectional Study"

_healthcare, 2024, doi:10.3390/healthcare12090906_

Round 1

Reviewer 1 Report

Comments and Suggestions for Authors

The authors conducted a very interesting investigation on the factors associated with Evidence-Based Practice Competencies among Taiwanese Nurses. The contents are relevant to nursing care.  However, the reporting should be improved, particularly in the discussion and conclusion sections.  I recommend that the authors make it explicit in the manuscript what this study adds to existing evidence and describe the findings more in-depth. It appears that the authors, in their discussions and findings, lost the study’s aim (utilizing the Taipei Evidence-Based Practice Questionnaire to assess the factors) and the added value it provided when compared to the available evidence. Please find attached my specific comments. 

Manuscript ID: healthcare-2956027

Title: Clear and informative.

Introduction

This section is very well described and structured following a logical flow. The contents are pertinent; however, I have some suggestions:

-          I recommend that the authors include a paragraph discussing EBP in the context of nursing care to emphasize the contents on the population target (i.e., why EBP are relevant in nursing practice) just after the second paragraph.

-          I suggest that the authors include two sentences that justify the appropriateness of the study’s design in the paragraph relating to the study’s rationale (i.e., in the last paragraph at line 64 just before the following sentence: “This study, therefore, aimed to investigate the 64 factors associated with EBP competencies in the Taiwanese healthcare context using the 65 comprehensive Taipei Evidence-Based Practice Questionnaire (TEBPQ)”. The authors can also decide to include this content in the study’s design section just after the declaration of the study’s design. If they decide to do so, I recommend that the authors separate the study design and data collection sections.

Methods

Overall, the methods are well presented. However, I have some suggestions below.

A clear description of the hospital setting is recommended to better interpret the study’s findings (i.e., private or public hospital system, hospital’s accreditations) since the context and healthcare systems could influence the acquisition of EBP nursing competencies. Please add a section that describes the hospital setting and characteristics.

It is unclear how the data was collected. Please add a section that clearly describes the study procedures, including questionnaire administration procedures and how the participants were informed about the study contents.

The authors should describe how they guarantee the confidentiality and anonymity of personal data. In this regard, I suggest the authors add a separate section (i.e., Ethical Considerations) that includes all information related to ethical issues. Therefore, the authors should move the contents related to ethical approval in this section.

Results

It is unclear what unit of measurement the data were presented in Table 2. Some categorical variables are expressed as mean and standard deviation rather than percentage, as is correctly stated in the text. If the data represent the average response scores for each item, both the table and the text should clearly indicate this.

Discussion

The structure is fine as well as the logical flow and presentation. However, each finding must be discussed in light of the available evidence from the literature. Some statements have no bibliographical references. Further, I suggest the authors start this paragraph by summarizing the study's rationale and novelty.

I suggest the authors not present the associations by listing them by number. It would be preferable to choose a more fluid and flowing arrangement following a logical sequence that is pertinent to the results (as the authors did).

The statements presented for debate on the association between TEBPQ scores and the work unit appear a little excessively hazardous and without evidence (lines 187-195) as every specialty is unique and distinct in terms of competencies and health needs. In addition, no bibliographic reference was associated to justify the hypothesis reported in lines 192-195. This fact is most likely more responsive to context-specific influences. Because this is a single-centre study, it is difficult to make inferences about this relationship. The causes for this significance should be examined in terms of unique local context characteristics.

Lines 197-201: no bibliographical reference was reported.

Lines 207-211: no bibliographical reference was reported.

Lines 219-227: no bibliographical reference was reported.

Please review the entire discussion section; every sentence should be supported by a literature reference if it is not part of the study results.

Conclusion

This section is not informative. The authors should start this section by summarizing the study's rationale and novelty along with the main results. Please add a paragraph summarising the study rationale, novelty, aim and the major findings.

Comments on the Quality of English Language

N.A.

Author Response

Reviewer 1, comment no. 1:

The authors conducted a very interesting investigation on the factors associated with Evidence-Based Practice Competencies among Taiwanese Nurses. The contents are relevant to nursing care.  However, the reporting should be improved, particularly in the discussion and conclusion sections.  I recommend that the authors make it explicit in the manuscript what this study adds to existing evidence and describe the findings more in-depth. It appears that the authors, in their discussions and findings, lost the study’s aim (utilizing the Taipei Evidence-Based Practice Questionnaire to assess the factors) and the added value it provided when compared to the available evidence. Please find attached my specific comments.

Response to Reviewer 1, comment no. 1:

We appreciate the reviewer’s thoughtful review and suggestions on improving the clarity and depth of the discussion and conclusion sections of our manuscript. We have made revisions to ensure that the aim of the study and its contribution to the existing evidence are explicitly stated and discussed. The following are our responses to each comment.

-----------------------------------------------------------------------------------------

Reviewer 1, comment no. 2:

Introduction

This section is very well described and structured following a logical flow. The contents are pertinent; however, I have some suggestions:

I recommend that the authors include a paragraph discussing EBP in the context of nursing care to emphasize the contents on the population target (i.e., why EBP are relevant in nursing practice) just after the second paragraph.

Response to Reviewer 1, comment no. 2:

We appreciate the reviewer’s suggestion and have added the following paragraph to the Introduction section: “In the context of nursing care, incorporating EBP into nursing education and practice is crucial for advancing nursing science, enhancing nursing care, and im-proving patient outcomes [8,9]. By combining the best available evidence with clinical experience and patient preferences, nurses are equipped to make well-informed decisions that optimize patient care. This approach enables the development of care protocols that are not only based on the latest and most reliable research but also addresses the unique needs of individual patients. As nurses continually adapt their practices to reflect new evidence, they can sustain ongoing professional development and quality improvement in healthcare settings [10].” [line 42 to 50]

-----------------------------------------------------------------------------------------

Reviewer 1, comment no. 3:

I suggest that the authors include two sentences that justify the appropriateness of the study’s design in the paragraph relating to the study’s rationale (i.e., in the last paragraph at line 64 just before the following sentence: “This study, therefore, aimed to investigate the 64 factors associated with EBP competencies in the Taiwanese healthcare context using the 65 comprehensive Taipei Evidence-Based Practice Questionnaire (TEBPQ)”. The authors can also decide to include this content in the study’s design section just after the declaration of the study’s design. If they decide to do so, I recommend that the authors separate the study design and data collection sections.

Response to Reviewer 1, comment no. 3:

In accordance with the reviewer’s suggestion, we have added the following sentences to the end of the Introduction section to enhance the rationale paragraph, as follows: “A cross-sectional study design was used for this broader exploration to assess these di-verse competencies and to provide an overview of the current state of EBP competencies in a specific healthcare setting.” [line 81 to 83]

-----------------------------------------------------------------------------------------

Reviewer 1, comment no. 4:

Methods

Overall, the methods are well presented. However, I have some suggestions below.

A clear description of the hospital setting is recommended to better interpret the study’s findings (i.e., private or public hospital system, hospital’s accreditations) since the context and healthcare systems could influence the acquisition of EBP nursing competencies. Please add a section that describes the hospital setting and characteristics.

Response to Reviewer 1, comment no. 4:

We thank the reviewer’s comment and has added the following description to the Section 2.1: “The hospital, a private hospital affiliated with the Buddhist Tzu Chi Foundation, is ac-credited by the Joint Commission of Taiwan. This accreditation body in healthcare was founded by the former Department of Health (renamed to Ministry of Health and Welfare in 2013), Taiwan Hospital Association, Taiwan Non-Government Hospitals and Clinics Association, and Taiwan Medical Association. The hospital has more than 500 beds and serves as a primary clinical education site for nursing students in the region”. [line 90 to 96]

-----------------------------------------------------------------------------------------

Reviewer 1, comment no. 5:

It is unclear how the data was collected. Please add a section that clearly describes the study procedures, including questionnaire administration procedures and how the participants were informed about the study contents.

Response to Reviewer 1, comment no. 5:

We appreciate the reviewer’s suggestion and have added description of the questionnaire administration procedures and how the participants were informed about the study contents, as follows: “A purposive sampling strategy was used to recruit participants. First, the head nurses from various units within the study hospital were approached and provided with a detailed explanation of the study. Following these briefings, they were given a number of survey packs, each containing a self-administered questionnaire, a consent form, and a convenience store gift card valued at 100 New Taiwan Dollars (approximately US$3.3). The head nurses then distributed these survey packs to the nurses in their respective units. Nurses who consented to participate completed the questionnaire and kept the gift card. Those who chose not to participate returned the uncompleted questionnaire with the gift card. Regardless of their decision, all were instructed to return the sealed survey packs to the head nurses. The researcher made regular visits to each unit to collect the returned survey packs from the head nurses.” [line 98 to 108]

-----------------------------------------------------------------------------------------

Reviewer 1, comment no. 6:

The authors should describe how they guarantee the confidentiality and anonymity of personal data. In this regard, I suggest the authors add a separate section (i.e., Ethical Considerations) that includes all information related to ethical issues. Therefore, the authors should move the contents related to ethical approval in this section.

Response to Reviewer 1, comment no. 6:

In response to the reviewer’s recommendations, we have added a new "Ethical Considerations" section to address the ethical issues associated with our study, as follows: “The survey packs were designed to be sealed by participants after completion, ensuring that their responses remained confidential and were only accessible to the research team. Upon collection, the questionnaire and consent forms were immediately stored separately to further safeguard participant anonymity. Each questionnaire was assigned a unique code, and to enhance privacy, no personal identifiers were included on any of the questionnaires. The study protocol received approval from the Research Ethics Committee of Dalin Tzu Chi Hospital, Buddhist Tzu Chi Medical Foundation (No. B11102009) (Approval date: May 11, 2022).” [line 111 to 118]

-----------------------------------------------------------------------------------------

Reviewer 1, comment no. 7:

Results

It is unclear what unit of measurement the data were presented in Table 2. Some categorical variables are expressed as mean and standard deviation rather than percentage, as is correctly stated in the text. If the data represent the average response scores for each item, both the table and the text should clearly indicate this.

Response to Reviewer 1, comment no. 7:

We thank the reviewer for pointing out the discrepancy in the units of measurement reported in Table 2. We apologize for the oversight regarding the presentation of categorical variables and have corrected the corresponding row header.

-----------------------------------------------------------------------------------------

Reviewer 1, comment no. 8:

The structure is fine as well as the logical flow and presentation. However, each finding must be discussed in light of the available evidence from the literature. Some statements have no bibliographical references. Further, I suggest the authors start this paragraph by summarizing the study's rationale and novelty.

I suggest the authors not present the associations by listing them by number. It would be preferable to choose a more fluid and flowing arrangement following a logical sequence that is pertinent to the results (as the authors did).

Response to Reviewer 1, comment no. 8:

We appreciate the reviewer for the constructive comments. We have adjusted our language to acknowledge more explicitly that our explanation is only a possibility. Further research is needed to confirmed our hypothesis.

We appreciate the recommendation to present the associations in a fluid narrative rather than as a numbered list. However, we would like to explain our rationale for maintaining the current format. Our analysis identified ten significant factors associated with the TEBPQ scores. Given the complexity and number of these factors, we believe that a numbered list makes it easier for readers to follow and reference specific factors clearly.

In addition, the order of these factors is not arbitrary but is organized based on the standardized regression coefficients. This means that factors with the largest effect sizes are discussed first, providing a logical sequence that reflects their relative impact on the TEBPQ scores.

We hope that with this explanation, the reviewer will agree that we can retain the existing structured, which can help with the comprehensibility of our findings, especially given the multiple variables discussed.

-----------------------------------------------------------------------------------------

Reviewer 1, comment no. 9:

The statements presented for debate on the association between TEBPQ scores and the work unit appear a little excessively hazardous and without evidence (lines 187-195) as every specialty is unique and distinct in terms of competencies and health needs. In addition, no bibliographic reference was associated to justify the hypothesis reported in lines 192-195. This fact is most likely more responsive to context-specific influences. Because this is a single-centre study, it is difficult to make inferences about this relationship. The causes for this significance should be examined in terms of unique local context characteristics.

Lines 197-201: no bibliographical reference was reported.

Lines 207-211: no bibliographical reference was reported.

Lines 219-227: no bibliographical reference was reported.

Please review the entire discussion section; every sentence should be supported by a literature reference if it is not part of the study results.

Response to Reviewer 1, comment no. 9:

We thank the reviewer for the constructive critique. We acknowledge the concerns raised about the potential overreach of our statements. We now clarify that while our findings suggest a possible link between work units and TEBPQ scores, such correlations should be interpreted with caution due to the unique competencies and health needs of different specialties, as well as the localized nature of our research context. Furthermore, we have adjusted our language to acknowledge more explicitly that these findings are preliminary and that further research is required to explore these dynamics in a broader, more diverse healthcare environment. The revised text is as follows: “It is hypothesized that this elevated risk might enhance the incentive among providers in these units to develop a heightened awareness and appreciation for EBP, potentially leading to stronger beliefs and more robust implementation of EBP. However, this hypothesis is preliminary and further research is needed to fully explore these dynamics in a broader healthcare context.” [line 231 to 235]

In response to “Lines 197-201: no bibliographical reference was reported.”, we have added the following sentence to the end of the paragraph: “However, further research is needed to substantiate this hypothesis, possibly by examining the direct impacts of various database use on TEBPQ scores or exploring longitudinal relationships between database usage and ongoing professional development in EBP.” [line 241 to 244]

In response to “Lines 207-211: no bibliographical reference was reported.”, we have revised the paragraph as follows: “It is hypothesized that younger nurses are likely to have graduated more recently and, therefore, may have been exposed to current EBP curricula, including EBP, leading to higher TEBPQ scores. In addition, it is possible that early-career professionals may engage more actively with EBP activities, workshops, and continued education, which can positively impact their TEBPQ scores. This conjecture about age-related disparities in EBP attitudes will require further research to confirm.”

In response to “Lines 219-227: no bibliographical reference was reported.”, we have added back the omitted reference, which is “Gauci, P.; Luck, L.; O'Reilly, K.; Peters, K. Workplace gender discrimination in the nursing workforce-An integrative review. J. Clin. Nurs. 2023, 32, 5693–5711.” [line 248 to 254]

-----------------------------------------------------------------------------------------

Reviewer 1, comment no. 10:

This section is not informative. The authors should start this section by summarizing the study's rationale and novelty along with the main results. Please add a paragraph summarising the study rationale, novelty, aim and the major findings.

Response to Reviewer 1, comment no. 10:

We appreciate the reviewer’s suggestion and have revised the Conclusion section as follows: “The findings of this study advance our understanding of EBP by integrating demographic and professional variables with EBP competencies, as quantified through the TEBPQ. Our findings indicate that younger professionals, males, registered nurses, and those in administrative roles exhibit higher EBP competencies, suggesting a generational shift and role-specific engagement in EBP. This could reflect differences in re-source access, responsibilities, or professional cultures.

On a practical level, this study supports the importance of targeted professional development and strategic resource allocation to enhance EBP competencies. The correlations between EBP training, proficiency in English, and database usage with higher competency scores also support the need for continuous professional development and more focused resource distribution, ensuring all staff are equipped for high-quality evidence-based care.” [line 304 to 315]

----------------------------------------------------------------------------------------------------------------------------------------------------------------------------------

Reviewer 2 Report

Comments and Suggestions for Authors

Thank you for the opportunity to review the article " Factors Associated with Evidence-Based Practice Competencies 2 among Taiwanese Nurses: A Cross-sectional Study ". Scientific-instrumental activity based on scientific evidence in the nursing profession is a key element of work, especially in modern times dominated by the rapid development of methods and standards of post-treatment in various clinical situations. Knowledge of procedures and methods of procedure results in professional care for the patient. The work needs to be supplemented by several issues:

line 73- A convenience sampling method was used to recruit 73 participants - what method? Please clarify;

tab3 - there is no calculation for the women's group;

Evidence-based courses or training - are there courses that do not take into account these reports?

line170-what factors were identified that influenced EBP competencies,

line 171 - most significant - what ? please specify;

Author Response

Reviewer 2, comment no. 1:

Thank you for the opportunity to review the article " Factors Associated with Evidence-Based Practice Competencies 2 among Taiwanese Nurses: A Cross-sectional Study ". Scientific-instrumental activity based on scientific evidence in the nursing profession is a key element of work, especially in modern times dominated by the rapid development of methods and standards of post-treatment in various clinical situations. Knowledge of procedures and methods of procedure results in professional care for the patient. The work needs to be supplemented by several issues:

line 73- A convenience sampling method was used to recruit 73 participants - what method? Please clarify;

Response to Reviewer 2, comment no. 1:

We appreciate the reviewer’s suggestion and have revised the description of the sampling method, as follows: “A purposive sampling strategy was used to recruit participants. First, the head nurses from various units within the study hospital were approached and provided with a detailed explanation of the study. Following these briefings, they were given a number of survey packs, each containing a self-administered questionnaire, a consent form, and a convenience store gift card valued at 100 New Taiwan Dollars (approximately US$3.3). The head nurses then distributed these survey packs to the nurses in their respective units. Nurses who consented to participate completed the questionnaire and kept the gift card. Those who chose not to participate returned the uncompleted questionnaire with the gift card. Regardless of their decision, all were instructed to return the sealed survey packs to the head nurses. The researcher made regular visits to each unit to collect the returned survey packs from the head nurses.” [line 98 to 108]

----------------------------------------------------------------------------------------- 

Reviewer 2, comment no. 2:

tab3 - there is no calculation for the women's group;

Response to Reviewer 2, comment no. 2:

We thank the reviewer for noting our omission in the variable label. The women’s group was used as the reference category. We have revised the variable label as “Male (ref. = female).

----------------------------------------------------------------------------------------- 

Reviewer 2, comment no. 3:

Evidence-based courses or training - are there courses that do not take into account these reports?

Response to Reviewer 2, comment no. 3:

In our manuscript, when we refer to "evidence-based courses or training," we specifically mean educational programs that are explicitly designed around the principles of evidence-based practice. These courses integrate the latest research findings and clinical evidence into the curriculum to enhance the knowledge and skills of healthcare professionals in utilizing evidence-based methodologies effectively. Indeed, there are courses that might not explicitly incorporate evidence-based reports or principles. However, we did not collect information on such courses from our participants and, therefore, are unable to analyze this aspect further.

-----------------------------------------------------------------------------------------

Reviewer 2, comment no. 4:

line170-what factors were identified that influenced EBP competencies,

Response to Reviewer 2, comment no. 4:

We thank the reviewer for inquiring the factors identified in our study. We have revised the sentence to “Ten factors significantly influencing EBP competencies were identified.”. In our study, we have identified 10 key factors associated with EBP competencies among Taiwanese nurses. These factors are listed in Table 3 of the manuscript. Furthermore, we have extensively revised the Discussion section to discuss each of these factors with findings from other studies where applicable. [line 208]

-----------------------------------------------------------------------------------------

Reviewer 2, comment no. 5:

line 171 - most significant - what? please specify;

Response to Reviewer 2, comment no. 5:

In response to the reviewer’s comment, we have revised the sentence to “Of them, the strongest association, as indicated by the magnitude of the standardized regression coefficients, was observed for those who had attended evidence-based courses or training in the past year.” This modification clarifies the basis for determining the strength of the associations observed in our study.

We used the magnitude of the standardized regression coefficient as a measure of the effect size in our regression analysis. A larger absolute value of the beta coefficient means a stronger effect. [line 209 to 211]

----------------------------------------------------------------------------------------------------------------------------------------------------------------------------------

Reviewer 3 Report

Comments and Suggestions for Authors

Dear authors

The manuscript presents an interesting topic. However, the authors still have a long way to go to publish this manuscript. I present my main concerns:

1- Introduction

- The introduction section should be thought of as the “visiting card” of the study, making the purpose and contribution of the investigation clear. However, in its current state, the introduction does not serve its purpose given that it is not entirely clear what this work adds to the literature, why it is essential, what is already known in general terms, and how authors intend to contribute.

- Authors should present TEBPQ definitions and relevance for healthcare professionals.

- The authors mention the factors associated with the TEBPQ, however they do not substantiate the factors presented.

Results

-        The participant descriptive statistics described in the results section should be moved to session 2.1.

-        The relevance of Figure 1 is unclear. A table with the descriptive statistics of the variables (mean, standard deviation, minimum and maximum values) would be sufficient to understand the results.

-        The authors should explain the relevance of including information about previous experience in evidence-based training.

-        The presentation of scale items in study results is not a usual procedure, authors should consider the possibility of removing the table with scale items. Alternatively, in the instruments section, authors may present examples of the items.

-        The authors mention the following phrase: "Table 3 presents findings from simple and multiple stepwise linear regression analyses, identifying factors associated with TEBPQ scores".

It is necessary to present a theoretical framework to support the 14 variables significantly associated with TEBPQ scores. One of the ways to substantiate these relationships is to define the study hypotheses. On the other hand, it was unclear the relevance of checking the factors that are associated with TEBPQ scores.

-        The authors collected data from employees and managers. Meaning that it is a multilevel study (individual and organizational levels). Therefore, the methodology used does not consider the levels of analysis.

-        The authors did not present correlations between the study variables.

-        It would be interesting to carry out confirmatory factor analysis for TEBPQ scale. It is not enough to mention that TEBPQ is an easy-to-use instrument with good validity and reliability for evaluating the effectiveness of EBP education.

DISCUSSION

-        The factors that affect TEBPQ scores should be substantiated. For example: How important is it to have adequate ability to read academic articles in English?

-        The relevance of studying TEBPQ scores should be duly substantiated. Furthermore, the relevance of TEBPQ scores for healthcare professionals should be substantiated.

-        The authors did not present the theoretical and practical contributions of the study.

Comments on the Quality of English Language

 Minor editing of English language required

Author Response

Reviewer 3, comment no. 1:

The introduction section should be thought of as the “visiting card” of the study, making the purpose and contribution of the investigation clear. However, in its current state, the introduction does not serve its purpose given that it is not entirely clear what this work adds to the literature, why it is essential, what is already known in general terms, and how authors intend to contribute.

Response to Reviewer 3, comment no. 1:

We thank the reviewer for the suggestion and have added the following to the Introduction section: “This study used the Taipei Evidence-Based Practice Questionnaire (TEBPQ), which was developed because of the recognized need for a succinct tool to assess healthcare professionals’ understanding and application of EBP principles in clinical settings [12]. The TEBPQ defines the EBP essential skills across five distinct domains: Ask, Acquire, Appraisal, Apply, and Attitude. These categories represent a systematic approach that mirrors the four-step EBP model at the clinical level, which involves formulating clin-ically relevant questions (Ask), gathering evidence (Acquire), evaluating its validity and applicability (Appraisal), and utilizing this evidence in clinical practice (Apply). In addition, the Attitude domain explores the psychological factors influencing EBP, evaluating practitioners' readiness and their perceived value of EBP. This study aimed to investigate the factors associated with EBP competencies in the Taiwanese healthcare context using the TEBPQ. A cross-sectional study design was used for this broader exploration to assess these diverse competencies and to provide an overview of the current state of EBP competencies in a specific healthcare setting.” [line 70 to 83]

-----------------------------------------------------------------------------------------Reviewer 3, comment no. 2:

Authors should present TEBPQ definitions and relevance for healthcare professionals.

Response to Reviewer 3, comment no. 2:

The TEBPQ is an instrument designed to assess EBP competencies across five key domains: Ask, Acquire, Appraisal, Apply, and Attitude. We have added the following text in the Introduction section: “This study used the Taipei Evidence-Based Practice Questionnaire (TEBPQ), which was developed because of the recognized need for a succinct tool to assess healthcare professionals’ understanding and application of EBP principles in clinical settings [15]. The TEBPQ defines the EBP essential skills across five distinct domains: Ask, Acquire, Appraisal, Apply, and Attitude. These categories represent a systematic approach that mirrors the four-step EBP model at the clinical level, which involves formulating clin-ically relevant questions (Ask), gathering evidence (Acquire), evaluating its validity and applicability (Appraisal), and utilizing this evidence in clinical practice (Apply). In addition, the Attitude domain explores the psychological factors influencing EBP, evaluating practitioners' readiness and their perceived value of EBP.” [line 70 to 79]

For healthcare professionals, the TEBPQ offers a metric to gauge their proficiency in these fundamental areas. It can support professional development and ensure that patient care is grounded in the best available evidence, ultimately improving outcomes.

-----------------------------------------------------------------------------------------Reviewer 3, comment no. 3:

The authors mention the factors associated with the TEBPQ, however they do not substantiate the factors presented.

Response to Reviewer 3, comment no. 3:

In our study, 10 factors were found to be significantly associated with TEBPQ scores, indicating a strong alignment with EBP competencies. The factors were identified using multiple stepwise regression analysis. We have discussed each of these factors in the Discussion section of our manuscript, offering possible reasons for their association with TEBPQ scores.

-----------------------------------------------------------------------------------------

Reviewer 3, comment no. 4:

Results- The participant descriptive statistics described in the results section should be moved to session 2.1.

Response to Reviewer 3, comment no. 4:

We thank the reviewer for the suggestion to move the participant descriptive statistics to Section 2.1. However, Section 2.1 (Study Design and Data Collection) is primarily focuses on the methodology rather than participant details. Instead, we have moved the sentence describing the response rate from Section 2.1 to the first line of the Results section. We hope this adjustment meets the journal's standards and addresses your concerns effectively.

-----------------------------------------------------------------------------------------

 Reviewer 3, comment no. 1:

The relevance of Figure 1 is unclear. A table with the descriptive statistics of the variables (mean, standard deviation, minimum and maximum values) would be sufficient to understand the results.

Response to Reviewer 3, comment no. 1:

We are grateful for the reviewer’s suggestion. Figure 1 was intended to offer a visual depiction of our data, aiding in the comprehension of overarching trends and patterns. In addition, as the response scale of the TEBPQ is a 5-point Likert-type scale, the data scale is ordinal rather than interval or ratio. This implies that the distances between response categories may not be equal, making the use of mean as a measure of central tendency potentially misleading. We hope that the reviewer will concur with our rationale for retaining Figure 1 in our presentation of results.

-----------------------------------------------------------------------------------------

Reviewer 3, comment no. 5:

The authors should explain the relevance of including information about previous experience in evidence-based training.

Response to Reviewer 3, comment no. 5:

We are grateful for the reviewer's comment regarding the relevance of including information about previous experience in evidence-based training. Given the potential variability in participants’ past exposure to such training, incorporating this factor into our multiple regression model enables us to account for its influence on EBP competencies more accurately. This adjustment ensures a more robust analysis of the factors contributing to EBP competencies.

-----------------------------------------------------------------------------------------

Reviewer 3, comment no. 6:

The presentation of scale items in study results is not a usual procedure, authors should consider the possibility of removing the table with scale items. Alternatively, in the instruments section, authors may present examples of the items.

Response to Reviewer 3, comment no. 6:

We appreciate the reviewer’s feedback regarding the presentation of scale items in our study results. However, we believe that including the scale items in the results section serves to facilitate readers' understanding of the content of the TEBPQ, thus enhancing the transparency of our findings. We hope that the reviewer will agree with our reason for retaining the scale items in the results section. However, we are open to the suggestion to relocate this information to the instrument section.

-----------------------------------------------------------------------------------------

Reviewer 3, comment no. 7:

The authors mention the following phrase: "Table 3 presents findings from simple and multiple stepwise linear regression analyses, identifying factors associated with TEBPQ scores".

It is necessary to present a theoretical framework to support the 14 variables significantly associated with TEBPQ scores. One of the ways to substantiate these relationships is to define the study hypotheses. On the other hand, it was unclear the relevance of checking the factors that are associated with TEBPQ scores.

Response to Reviewer 3, comment no. 7:

We appreciate the reviewer’s comment. In our study, the development of the questionnaire and the choice of variables investigated were indeed based on review and synthesis of previous research in EBP. Our primary objective was to empirically explore the association of various factors with EBP competencies as measured by the TEBPQ, rather than testing these associations within a predefined theoretical framework. We acknowledge the value of a strong theoretical basis for research studies, as it enhances the understanding and interpretation of the findings. However, for this particular study, our aim was to conduct a broad exploratory analysis that could surface new insights and patterns without the constraints imposed by a specific theoretical framework.

-----------------------------------------------------------------------------------------

Reviewer 3, comment no. 8:

The authors collected data from employees and managers. Meaning that it is a multilevel study (individual and organizational levels). Therefore, the methodology used does not consider the levels of analysis.

Response to Reviewer 3, comment no. 8:

We agree that a multilevel analysis could provide additional insights into the interplay between individual and organizational factors. However, for the scope of this study, we chose to concentrate on the individual level to maintain the focus of our analysis. We believe this approach still allows us to draw meaningful conclusions about EBP competencies.

-----------------------------------------------------------------------------------------

Reviewer 3, comment no. 9:

The authors did not present correlations between the study variables.

Response to Reviewer 3, comment no. 9:

We thank the reviewer’s comment regarding the presentation of correlations between the study variables. However, our study objective does not prioritize the examination of correlations among our factors. Instead, our focus lies on assessing the independent associations between each independent variable and the Taipei Evidence-Based Practice Questionnaire (TEBPQ). Nevertheless, we have evaluated the presence of multicollinearity among the independent variables using the variance inflation factor (VIF). We believe that by addressing multicollinearity, we have taken appropriate measures to safeguard the integrity of our regression analysis.

-----------------------------------------------------------------------------------------

Reviewer 3, comment no. 10:

It would be interesting to carry out confirmatory factor analysis for TEBPQ scale. It is not enough to mention that TEBPQ is an easy-to-use instrument with good validity and reliability for evaluating the effectiveness of EBP education.

Response to Reviewer 3, comment no. 10:

We greatly appreciate the reviewer’s invaluable suggestion regarding the use of confirmatory factor analysis for the TEBPQ scale. We agree that implementing a CFA would provide a robust test of the scale’s factor structure and further validate its use in assessing the effectiveness of evidence-based practice education. As such, we plan to conduct a CFA as part of a separate, dedicated study. This will allow us to comprehensively explore the psychometric properties of the TEBPQ in greater detail.

Regarding the current study, the statement concerning the TEBPQ’s validity and reliability was based on findings from the developer’s original study. In that research, the TEBPQ demonstrated strong psychometric properties, which encouraged its use in our research.

-----------------------------------------------------------------------------------------

Reviewer 3, comment no. 11:

The factors that affect TEBPQ scores should be substantiated. For example: How important is it to have adequate ability to read academic articles in English?

Response to Reviewer 3, comment no. 11:

We than the reviewer for the comment. The ability to read academic articles in English is a significant factor that can influence the scores with a standardized regression coefficient of 0.088 (p = 0.005). English is the lingua franca of scientific literature, and a substantial portion of research articles are published in this language. Therefore, having a proficient level of English reading ability is crucial for healthcare professionals to access, understand, and apply the latest evidence in their practice.

Nevertheless, we did not directly measure the participants’ English reading ability in our study. We appreciate your suggestion and will consider incorporating an objective measure of English proficiency in our future studies to better understand its impact on EBP competen

-----------------------------------------------------------------------------------------

Reviewer 3, comment no. 12

The relevance of studying TEBPQ scores should be duly substantiated. Furthermore, the relevance of TEBPQ scores for healthcare professionals should be substantiated.

Response to Reviewer 3, comment no. 12:

We appreciate the reviewer’s feedback emphasizing the need to substantiate the relevance of TEBPQ scores, particularly for healthcare professionals. The development of the TEBPQ was motivated by the recognized necessity for a succinct tool to measure the self-efficacy of EBP and outcomes among healthcare workers. This instrument assesses how well healthcare professionals understand and apply EBP principles. By assessing these scores, healthcare institutions can ensure that their staff adheres to the latest research and clinical guidelines, thereby maintaining high standards of patient care. Moreover, TEBPQ scores help identify specific areas lacking in EBP knowledge, guiding targeted educational initiatives or interventions to improve these competencies. Moreover, these scores offer a benchmark for organizational performance in implementing EBP. We have added the following statement to the end of the Introduction section: “This study used the Taipei Evidence-Based Practice Questionnaire (TEBPQ), which was developed because of the recognized need for a succinct tool to assess healthcare professionals’ understanding and application of EBP principles in clinical settings [15].” [line 70 to 73]

-----------------------------------------------------------------------------------------

Reviewer 3, comment no. 13

The authors did not present the theoretical and practical contributions of the study.

Response to Reviewer 3, comment no. 13:

We appreciate the thoughtful and constructive feedback. Our study contributes to the existing literature on EBP by using the TEBPQ to assess EBP competencies within the Taiwanese healthcare context. From a theoretical standpoint, this study advances our understanding by integrating demographic and professional variables with EBP competencies as measured by the TEBPQ. Our regression analysis explored how age, sex, and job roles influence EBP competency levels. For example, the inverse relationship between age and TEBPQ scores suggested a generational shift in attitudes towards EBP, suggesting that younger professionals may be more receptive to EBP methodologies. Moreover, the findings that being a male, a registered nurse, and holding administrative roles are associated with higher TEBPQ scores contribute to the theoretical understanding of role-specific engagement with EBP. This could reflect underlying factors, such as differential access to resources, varying responsibilities, or distinct professional cultures that influence EBP competency.

On a practical level, this study informs professional development, recruitment, policy-making, and resource allocation within healthcare settings to enhance EBP competencies. By identifying that younger nurses and those in specialized roles like gynecology or pediatrics exhibit higher EBP competencies, tailored training programs can be developed to capitalize on these strengths and address weaknesses in other groups. Furthermore, the correlation between higher EBP competencies with demographics such as male nurses and those in administrative positions can guide more strategic recruitment and role assignments to foster better EBP implementation. The strong link between EBP training participation and increased competencies underscores the need for supportive educational policies that ensure continuous professional development for all staff, especially those showing lower competencies. Furthermore, the association between proficiency in English and database usage are higher EBP skills allows for more targeted resource distribution, such as improved access to medical journals and database training.

In response to the reviewer’s suggestion, we have revised our Conclusions to the following: “The findings of this study advance our understanding of EBP by integrating demographic and professional variables with EBP competencies, as quantified through the TEBPQ. Our findings indicate that younger professionals, males, registered nurses, and those in administrative roles exhibit higher EBP competencies, suggesting a generational shift and role-specific engagement in EBP. This could reflect differences in re-source access, responsibilities, or professional cultures.

On a practical level, this study supports the importance of targeted professional development and strategic resource allocation to enhance EBP competencies. The cor-relations between EBP training, proficiency in English, and database usage with higher competency scores also support the need for continuous professional development and more focused resource distribution, ensuring all staff are equipped for high-quality evidence-based care.” [line 304 to 315]

-----------------------------------------------------------------------------------------

Round 2

Reviewer 1 Report

Comments and Suggestions for Authors

The authors have addressed all my comments. There are some spelling and grammar errors. Please, revise the quality of English language. 

Comments on the Quality of English Language

There are some spelling and grammar errors. Moderate editing of English language is required.

Author Response

We thank the reviewer for pointing out the spelling and grammar errors in our manuscript.  We have revised the manuscript to correct the identified spelling and grammatical mistakes. 
